# The Predictive Factors of Hospital Bankruptcy—An Exploratory Study

**DOI:** 10.3390/healthcare11020165

**Published:** 2023-01-05

**Authors:** Bradley Beauvais, Zo Ramamonjiarivelo, Jose Betancourt, John Cruz, Lawrence Fulton

**Affiliations:** 1School of Health Administration, Texas State University, Encino Hall, Room 250A, 601 University Drive, San Marcos, TX 78666, USA; 2Boston College, Woods College of Advancing Studies, St. Mary’s Hall South, Chestnut Hill, MA 02467, USA

**Keywords:** hospital, bankruptcy, financial distress, forecast, predictive models

## Abstract

The United States healthcare industry has witnessed a number of hospitals declare bankruptcy. This has a meaningful impact on local communities with vast implications on access, cost, and quality of care available. In our research, we seek to determine what contemporary structural and operational factors influence a bankruptcy outcome, and craft predictive models to guide healthcare leaders on how to best avoid bankruptcy in the future. In this exploratory study we performed, a single-year cross-sectional analysis of short-term acute care hospitals in the United States and subsequently developed three predictive models: logistic regression, a linear support vector machine (SVM) model with hinge function, and a perceptron neural network. Data sources include Definitive Healthcare and Becker’s Hospital Review 2019 report with 3121 observations of 32 variables with 27 observed bankruptcies. The three models consistently indicate that 18 variables have a significant impact on predicting hospital bankruptcy. Currently, there is limited literature concerning financial forecasting models and knowledge detailing the factors associated with hospital bankruptcy. By having tailored knowledge of predictive factors to establish a sound financial structure, healthcare institutions at large can be empowered to take proactive steps to avoid financial distress at the organizational level and ensure long-term financial viability.

## 1. Introduction

Bankruptcy is an unfortunate outcome for any business. The loss of invested capital, jobs, and product or service offerings in the local area can be abrupt and meaningful. However, when a hospital closes, the ramifications can have a far more detrimental impact on the entire community. The immediate deterioration of access to clinicians, diagnostic testing, and supportive clinical services can impose undue burdens on the patients and families who have relied on that particular hospital for their healthcare needs.

Unfortunately, in recent years, the closure of the local hospital has become an all-too-common occurrence across the country. The Polsinelli quarterly report pertaining to financial distress across industries noted that although bankruptcy filings in most industries have decreased by 53% nationwide since 2010, in healthcare, filings have increased by 305% in that same time frame [1]. Most of these filings have been for hospitals in rural settings. According to a recent report from the healthcare consulting company, Chartis Group, 138 rural hospitals have closed since 2010. Even before the COVID-19 pandemic, a staggering 44% of rural hospitals operated at a loss [2]. Since the pandemic began, rural facilities have been challenged significantly by the loss of revenue from more profitable diagnostic, outpatient, and ancillary services. This additional burden has simply proven too much for many smaller facilities to bear. Without the requisite financial reserves to draw upon to sustain operations, bankruptcy and closure have become the only available options.

Similar concerns of hospital bankruptcy exist in urban settings, particularly regarding safety net hospitals, which serve most of the nation’s urban poor. These facilities have struggled to remain solvent in recent years due to increased technology costs, declining governmental sponsored insurance reimbursement, inflationary staffing cost pressures, increased demand from the uninsured in states where Medicaid coverage was not expanded, intensified competition, and, more recently, loss of non-COVID related services [3]. Knowing the factors associated with bankruptcy may help hospitals pay more attention to these factors to avoid bankruptcy. The purpose of this study is to build a predictive model to forecast bankruptcy among US hospitals.

### Literature Review

Several studies have provided some financial tools to predict bankruptcy as evidenced by the scoping review of bankruptcy prediction from 1930 to 2007 [4]. Bellovary et al. (2007) and Jackson and Wood (2013) highlighted the evolution of bankruptcy prediction models [4,5]. The earliest models consist of univariate analyses of financial ratios [4,6,7,8,9]. These studies found the following ratios as good predictors of organizational failures: working capital/total assets, surplus and reserves/total assets, net worth/fixed assets, fixed assets/total assets, current ratio, net worth/total assets, sales/total assets, cash/total assets, current assets/total assets, net worth/total debts, and net profit/total assets [4,6,7,8,9].

Other studies compared financial ratios between bankrupt and non-bankrupt firms. For instance, Beaver (1966), after comparing the mean of 30 ratios of 79 failed and 79 non-failed organizations in 38 industries, suggested the following ratios to have the highest accuracy in bankruptcy prediction ability, one year before bankruptcy, net income/total debt, net income/net worth, cash flow/total debt, cash flow/total assets, and net income/sales [10]. However, Beaver (1966) and Altman (1968) suggested that predicting bankruptcy using univariate analysis of individual ratios may lead to faulty and ambiguous conclusions. For instance, an organization may have low profitability but above-average liquidity [10,11]. This situation does not allow us to draw a clear-cut determination of whether the organization is at risk of financial failure [11]. Additionally, since extant studies suggested different ratios, with different levels of importance, to predict bankruptcy, it is difficult to pinpoint which of these financial ratios are the best predictors of bankruptcy [11].

Therefore, Beaver (1966) and Altman (1968) suggested that using multivariate analyses to combine multiple ratios into a single predictive model is a better way to assess an organizations’ financial situation than using single financial ratios [10,11]. Edward Altman pioneered the use of the combination of financial ratios to predict bankruptcy. In his first study, (Altman, 1968) used a discriminant analysis of 33 publicly-traded bankrupt and 33 non-bankrupt manufacturing firms to extract a financial index, the Altman Z-score model, to predict bankruptcy. The Z-score model consists of the combination of weighted multiple financial ratios (working capital/total assets; retained earnings/total assets; earnings before interest and taxes/total assets; the market value of equity/book value of total debts; sales/total assets). From the discriminant analysis, Altman suggested a Z-score of 2.675 as the score dividing bankrupt and non-bankrupt firms [11]. Later on, based on a subsequent study of different samples, Altman indicated that 1.81 should be the cut-off Z-score that has a more predictive accuracy separating bankrupt and non-bankrupt firms than the former 2.675 Z-score. Altman also built two other Z-score models to predict bankruptcy among private-owned firms and non-manufacturing firms, the Z′-score and the Z″-score models, respectively [12].

Recently, statistical learning techniques such as logit, probit, and survival analysis along with machine learning techniques such as neural networks and recursive partitioning have become more prominent in bankruptcy studies [4,5,13,14]. These studies have focused on bankruptcies for any organization, not just hospitals. While instructive to a point, the non-healthcare context of these studies is, unto itself, a limitation given hospitals’ organizational and ownership differences, reimbursement variation, and overall societal impact.

Early studies of hospitals suggested financial and non-financial factors are associated with hospital financial distress. For example, extensive literature found negative cash flow, low cash flow/total debt, low current ratio, negative equity, and loss of profitability as indicators of hospital financial distress or bankruptcy [15,16]. Additionally, Bazzoli and Andes (1995) used Standard and Poor’s BBB credit ratings of newly issued bonds to distinguish between financially and non-financially distressed hospitals. Then, they compared a set of financial ratios of the distressed hospitals with those of the financially sound hospitals [17]. They found that distressed hospitals had fewer assets and negative profitability in terms of total margin and return on assets, as well as high debt and poor liquidity, compared with their financially sound counterparts [17]. Bankruptcy and financial distress were also associated with low occupancy rate, slow collection of account receivables, poor payer mix in terms of a high percentage of Medicare and Medicaid patients, and aging facilities [15,16,17], as well as poor management, fraud allegations, demographic changes, financial strategy/desire to sell, quality issues, physician malpractice insurance, physician politics and external politics [18]. Additionally, compared with hospitals that filed for bankruptcy but remained open, hospitals that filed for bankruptcy and permanently closed their doors were more likely to be for-profit hospitals and have lower cash flow/total debt ratio [17]. However, in nearly all cases, these studies were disadvantaged by relatively small sample sizes and nascent methodologies.

More robust methods have been used to attempt to apply bankruptcy predictive models, such as the Financial Strength Index [19], the Altman Z-score model [20,21,22,23,24], the Ohlson O Score [23,24,25], and the Zmijewski score [23] within studies on financial distress/failure of health care organizations. For instance, Puro et al. (2019) assessed the predictive accuracy of three models: the modified Altman Z-score, the Ohlson O score, and the Zmijewski score; they did not find a single ratio that consistently separates bankrupt and non-bankrupt hospitals across the three models [22]. In the same vein, Corbett and Gosset assessed the effectiveness of financial and non-financial variables in predicting the financial solvency of private for-profit hospitals in the US. They included predictive models including Altman Z-score, Altman Z-score_2, Financial Strength Index, and Financial Strength index_2, as well as financial ratios and non-financial variables in their studies [25]. They framed their studies based on cash flow theory, resource dependence theory, organizational-environmental theory, and Kissick’s iron triangle [25]. They found that none of the models, nor the financial and non-financial ratios, were significantly associated with for-profit hospital financial solvency. Similar to prior studies’ limitations, these authors suggested a small sample size as a possible reason for their findings [25].

Therefore, building on previous studies and their recognized limitations in contemporary methods, sample size, and applicability of broader industry findings, we seek to advance our understanding of the factors associated with hospital bankruptcy in the United States by (1) including and critically analyzing the current market and hospital characteristics and (2) leveraging advancements in analytical techniques. Our ultimate intent is to develop a predictive model to guide hospital and healthcare leaders toward understanding what specific contemporaneous factors and conditions are likely to financially imperil hospitals and lead to their declaration of bankruptcy.

## 2. Materials and Methods

### 2.1. Methods

Forecasting bankruptcy (a binary variable) is a classification problem that may be approached by statistical and machine-learning techniques. To be meaningful, however, the importance and directionality of each possible predictor must be assessed. To address this last requirement, we selected binary classification techniques that would produce parameter estimates with associated directionality and importance of the predictors including a logistic regression (LR) model, linear support vector machine (SVM-stochastic gradient descent with a hinge loss function) model, and perceptron (one-layer neural network-NN) model.

LR is an appropriate supervised classification technique for binary (dichotomous) dependent variables modeled as a function of predictors (fixed or random). Traditional regression cannot estimate the probability of the dependent variable given the independent variables; however, logistic regression solves this problem. Given a dichotomous dependent variable (*Y*), the logistic regression model is specified as shown in Equation (1) [26].
(1)$$P\left(Y\right)=\frac{e^{\beta_{0}+\beta_{1}X_{1}+\ldots \beta_{n}X_{n}}}{1+e^{\beta_{0}+\beta_{1}X_{1}+\ldots \beta_{n}X_{n}}}$$

In Equation (1), the right-hand side is a logistic function with Euler’s *e* raised to the power of the regression equation. By observation, one can see that the maximum value of the right-hand side approaches one for positive values of the traditional regression equation and zero for negative values. Manipulating Equation (1) produces a simpler equation for maximum likelihood estimation where the left-hand side is the log-odds (Equations (2) and (3)).
(2)$$\frac{P\left(Y\right)}{1-P\left(X\right)}=e^{\beta_{0}+\beta_{1}X_{1}+\ldots \beta_{n}X_{n}}$$
(3)$$log\left(\frac{P\left(Y\right)}{1-P\left(X\right)}\right)=\beta_{0}+\beta_{1}X_{1}+\ldots \beta_{n}X_{n}$$

In Equation (2), the right-hand side is the odds ratio (probability of bankruptcy divided by the probability of no bankruptcy). Applying the natural logarithm to both sides (Equation (3)) results in a regression-like equation for the log odds that may be estimated using maximum likelihood estimation. Another method for binary classification that produces both coefficient weights and directionality is SVM, a stochastic gradient descent (SGD) classifier with a hinge loss function. Instead of using traditional logistic regression approaches, this technique looks for the optimal separating hyperplane between the bankrupt and non-bankrupt categories. SGD, a gradient descent method that uses mini-batches rather than the entirety of the dataset at the same time, is used to optimize the hyperplane separation. If the loss function were a logistic function instead of a hinge function, then the results would be equivalent to logistic regression.

The hinge loss function for a dichotomous variable observation is shown in Equations (4) and (5) [27].
(4)$$L\left(y\right)=\max\left(0,\;1-y\;\cdot \;\hat{y}\right)$$
(5)$$L\left(y\right)=\max\left(0,\;1-y\;\cdot \left(b_{0}+b_{1}X_{1}+\ldots b_{n}X_{n}\right)\right)$$

In Equation (4), y is the actual dichotomous value (either −1 or 1), and $\hat{y}$ is the estimator produced during classification ($b_{0}+b_{1}X_{1}+\ldots b_{n}X_{n}$ for a linear SVM as shown in Equation (5)). When the classifier is correct (y and $\hat{y}$ have the same sign) and |$\hat{y}$| is greater than 1, the value of the loss function is 0, which means that it falls on the proper side of the separating hyperplane. However, if the classifier is correct but by not enough separating margin (e.g., |$\hat{y}$| < 1), then the loss function is positive. Further, when the classifier is not correct, the loss function will be positive as well. Minimizing the loss function then maximizes the separating hyperplane.

Figure 1 depicts a linear SVM with a hinge-loss function. In Figure 1, bankruptcies are pictured in green and non-bankruptcies in red. Losses are zero for those observations clearly outside of the separating hyperplane boundaries. They are non-zero for those within the boundary or misclassified. Linear SVM seeks the optimal separating hyperplane that minimizes the losses. The hinge loss function is convex, so traditional optimizers are available.

A third model available for classification that provides both directionality and magnitude information is the perceptron, the simplest neural network available. The perceptron takes scaled inputs, weights them (where the weights are tuned through optimization during backpropagation), aggregates them, and then feeds them to a Heaviside step function (i.e., 1 if $\hat{y}$ > 0, 0 otherwise). Since each input is associated with only a single weight (coefficient), estimates of both the strength and direction of association between the dependent and independent variables are available. Figure 2 depicts a perceptron.

In Figure 2, the standardized inputs (blue circle) are associated with weights (black lines with *w*) and aggregated (orange circle). This weighted combination is then processed through the Heaviside step function (red circle) and evaluated as either bankrupt or non-bankrupt by the step function (gray circle). This process is sometimes done in batches (e.g., 64 observations at a time), and the estimates (gray circle) are then compared to the true state of nature (green circle). The loss function for the batch is calculated (black circle), and weights are updated that would have minimized the loss function for that batch. Additional batches are then processed through the perceptron.

By using a collection of interpretable machine learning methods to estimate unseen, pristine test data, researchers can build models that provide coefficient directionality and effect size. These models can be compared for congruence based on coefficient magnitude and directionality. The better-performing models may be identified and even ensembled (e.g., averaging) to push them forward for forecasting. In healthcare, this approach is particularly necessary to ensure that specious results from a single model are avoided and to handle the ‘big data’ that exists within the sector.

### 2.2. Data

Data were obtained from both Definitive Healthcare and Becker’s Hospital Review (bankruptcy status). Definitive Healthcare contains the databases of several US healthcare organizations, such as hospitals, physician group practices, surgery centers, and long-term care organizations. With respect to US hospital data, Definitive Healthcare combines data from several sources, such as the American Hospital Association Annual Survey (hospital profile), Medicare Cost Report (financial data), and Hospital Compare (healthcare quality data). Definitive Healthcare provided 3222 observations with 34 variables from the year 2019. Overall, 2.4% of the observations were missing (2651 data points out of 109,548). Two variables (Hospital Compare Scores and the overall star rating from the Hospital Consumer Assessment of Health Care Providers and Systems (HCAHPS) and (separately) 101 observations contained more than 10% missing values. These were dropped from the analysis. None of the variables or observations were associated with a positive bankruptcy status. Medians were imputed for the remaining missing data (less than 1% of all observations). The final dimensionality of the data was 3121 observations of 32 variables. Federal hospital systems including the Veterans Affairs hospitals, the Indian Health Service, and the Military Health System were excluded from our study sample due to a lack of numerous data elements relevant to our study and because each of the hospitals in these systems is directly funded by the federal government.

### 2.3. Variables

The dependent variable for this study is the declaration of bankruptcy as reported in Becker’s Hospital Review for the year 2019. The year 2019 was specifically chosen as it is the most recent complete year of data devoid of any potential influence of the COVID-19 pandemic. Numerous independent variables are included in the study to account for the variation in hospital bankruptcy associated with various individual hospital and hospital market characteristics, including whether the hospital is an Academic Medical Center or not, hospital ownership type (for profit vs. not-for-profit), government-operated or not, whether the facility is a sole community hospital or not, local hospital market concentration (as measured via the Herfindahl–Hirschman Index), the hospital case mix index, the number of staffed beds in the facility, urban or rural location, Medicare percent payer mix, Medicaid percent payer mix, the hospital serious complication rate, the hospital bed utilization rate, Joint Commission accreditation, and the percent of patients that would recommend the hospital to friends and family (HCAHPS survey results). Numerous financial factors are also considered, including the current ratio, total assets, days cash on hand, net patient revenue, operating income per staffed bed, accounts receivable balance, uncompensated care as a percentage of net patient revenue, debt to equity ratio, net operating profit margin, the asset turnover ratio, and the hospital age of plant.

### 2.4. Analysis

Our preliminary analysis was conducted via logistic regression to evaluate the data and gain an initial perspective regarding the directionality and level of significance among the variables in our dataset. We then randomly selected a 50% test set which was separated from the data for use in prediction. Data were highly imbalanced with only 27 observations of bankrupt organizations in the set of 3121 observations, so it was necessary to use an even split between the training and test sets. To address the low number of bankruptcies available for training, the training set was augmented with 120 bootstraps of the bankrupt companies. The final structure of the training set was 3120 observations, whereas the test set was 1561 observations. The test set included 14 bankrupt observations. Standard scaling (Z-transformation) was fit to the training set and the saved parameters from the training set were then used to transform the test set to ensure no information leakage. This transformation supported non-tree-based methods which are not scale invariant. Due to the imbalanced data, the primary metrics of interest were precision (positive predictive value) and recall (sensitivity) as applied to the pristine test set. The F1 score, a weighted combination of precision and recall, as well as accuracy and specificity were evaluated in all cases. Where appropriate, parameter estimates, and feature importance were calculated to estimate the effects of the independent variables. Hyperparameter tuning for all models was conducted using the training set data. Final models were then used to forecast bankruptcy status on the test set. Analyses were conducted in Python 3.7 and are posted to GitHub. They are available at the website provided below: https://github.com/dustoff06/Bankruptcy/blob/main/Bankruptcy%20Final.ipynb.

## 3. Results

A descriptive analysis of all variables is available in Table 1.

For logistic regression on the training set (where the models were built), Variance Inflation Factors (VIF) were calculated. The highest VIF was 4.43 (Net Patient Revenue). Investigation of the model standard errors and coefficient directionality suggested that the logistic regression model provided stable and congruent results. Table 2 provides the traditional logistic regression analysis of odds ratios and *p*-values. Highlighted lines identify variables that are statistically significant. From this table, it is clear that Joint Commission Certification, a higher number of HCAHPS recommendations, and the level of a hospital’s accounts receivable are associated with lower likelihoods of bankruptcy. Conversely, a higher the percentage of Medicaid in the organizational payer mix, a higher facility age, and an elevated serious complication rate all significantly contribute to a higher likelihood of financial collapse.

In the ‘Coefficient (Se)’ column of Table 2, the values of the coefficient (showing directionality and magnitude of the relationship between bankruptcy and the predictor in the “Variable” column) are provided along with the standard error, which is offset with parentheses. The ‘Odds Ratio (95% CI)’ column provides both the odds ratio (exponent of the coefficient) along with a 95% confidence interval in parenthesis. The associated Gaussian standard normal value along with the related *p*-value (in parentheses) is in the ‘Z-value (*p*-value)’ column. As an example, the variable ‘TJC Certified’ has a strong negative relationship with bankruptcy as indicated by the coefficient estimate and the standard error. Converting the coefficient to odds ratios shows that the odds of bankruptcy to no bankruptcy are 0.028 to 1.000, much smaller than 1:1 under the null hypothesis. The value in the standard normal of the coefficient is −7.925 or nearly eight standard deviations below the mean. The *p*-value is therefore near zero indicating a statistically significant result.

Table 3 provides the metrics for all of the predictive models in our analysis. Overall accuracy ranged between 68 to 69%. All models achieved recalls (sensitivities) of 79%, classifying 11 of the 14 bankruptcies in the test set correctly. Models ranged from 67 to 69% specificity (recall for companies not bankrupt). In all cases, the models have low positive predictive values. The SGD model (for example) classified 11 of 14 bankrupt companies in the test set correctly but also classified 476 companies out of 1547 non-bankrupt as bankruptcies. Thus only 11 out of 487 predicted to be positive were actually positive. This can be explained by the fact that the training models used balanced data to identify those organizations with symptoms of bankruptcy. Those observations that are misclassified may reflect organizations which are struggling financially and have indicators pointing toward future bankruptcy.

Table 4 provides a coefficient comparison of all three models. Directionality differences are highlighted at the bottom of the table, and the directionally consistent variables are sorted by the absolute average of the coefficients. Net patient revenue, accounts receivable, current-ratio, total assets, debt-to-equity ratio, net operating profit margin, and uncompensated care by net patient revenue are all associated with the decreased classification of the bankrupt status. Similarly, TJC certification, government status, and HCAHPS recommendation are all also associated with the decreased classification of bankruptcy. Hospitals’ asset turnover ratio, a measure of a company’s revenues relative to its assets, is associated with a decreased bankruptcy classification in two of the three models. The next 16 variables sorted by the absolute value of the average of the coefficients are all directionally consistent.

## 4. Discussion

Although not perfectly consistent, all three of our predictive models point to several salient and significant factors that contribute to hospital bankruptcy. In our final analysis, we list a total of twenty-five contributory variables, but for the purposes of brevity, we profile the top ten on the list in the section below.

If we take a simple average of the absolute values of each factor across the three models, the most important factors in whether the hospital remains solvent are first, the facilities’ levels of ‘net patient revenue’ and second, the amount in ‘accounts receivable’. These results are logical because without revenue—either directly received or as a receivable—the organization soon ceases to exist. Although there are clearly important nuances to how quickly receivables can be turned into cash flow, the simple existence of receivables balances on the balance sheet indicates a positive level of performance that is more likely than not to facilitate long-term financial sustainability. Devoid of an opportunity to collect revenue—either directly or via a receivable—then the opportunity to generate cash flow ceases to exist. Further, given the specific characteristics of the third-party payment system in the United States, it is virtually impossible to avoid carrying a receivables balance of some sort. Thus, we infer that a hospital showing a higher receivables balance is more likely one that is meeting patient demand and generating revenue. A simple truth in all businesses is revenue must exceed expenses. Healthcare is not immune from this fact.

The third attribute on our list is whether or not the facility is ‘Joint Commission (TJC) accredited’. This came as somewhat of a surprise to us initially. However, as we examined the outcome more closely, the more it made sense. In our study, a TJC affiliation was identified versus other competing accreditation agencies including Det Norske Veritas and the Healthcare Facilities Accreditation Program. Clearly, there are inherent financial advantages to being accredited at all given the national recognition, clinical performance, and attractiveness to insurers and providers. However, the TJC ‘brand name’ recognition to insurers and levels of performance which TJC accredited facilities attain are factors which likely have an impact in our study.

Fourth on our list of factors is whether a facility is a ‘government-operated’ public hospital or not. We attribute this to the simple fact that government-owned and operated facilities generally have access to taxpayer support and lower costs of capital via municipal bonds and/or hospital revenue bonds which most private not-for-profit and for-profit organizations cannot utilize. We should also note that in our analysis, we excluded all Veterans Administration (VA), Indian Health Service (IHS), and Military Healthcare System (MHS) facilities from this group as they are indemnified against bankruptcy by the US taxpayer. Thus, the remaining public organizations tend to be larger academic medical centers such as UCLA (CA), NYC Health (NY), Harris Health (TX), and the Hennepin County Medical Center (MN) among numerous others. In addition to the support these facilities receive, which we referenced earlier, some of the organizations just listed also garner support by virtue of being academic medical centers. Such facilities garner not just increased reimbursement via the Medicare inpatient prospective payment system, but also are able to generate considerable support via grants, medical education funding from Medicare, and other research financing.

The fifth variable on our list aligns with reasonable expectations. The ‘current ratio’ is negatively associated with bankruptcy—inferring that the more liquidity that the hospital maintains, the less likely it is that it will go bankrupt. The higher this ratio reaches, it simply means that there are more cash and liquid assets (i.e., marketable securities, receivables, and inventory) available to meet short-term liabilities. Given the organization can turn less liquid assets such as receivables quickly into cash, it would make sense that there is less of a likelihood that bankruptcy becomes an immediate threat.

Sixth on our list is the ‘HCAHPS survey of whether patients would refer their family members or friends to the facility’. Although there is not an inherently clear linkage between patient perceptions of quality and financial outcomes, these findings are consistent with prior studies that confirm patient perceptions of quality are associated with hospital financial performance [28]. These studies show a positive patient experience is associated with increased profitability and a negative patient experience is even more strongly associated with decreased profitability [29].

The seventh and eighth variables on our list also logically align with our expectations. The level of an organization’s ‘total assets’ provides a cushion against the immediate financial threat as assets can be leveraged or sold to maintain liquidity and solvency. Likewise, the ‘debt-to-equity ratio’ implies that reasonable use of debt financing provides not just tax advantages, but also limits non-growth enhancing payouts for dividends to for-profit hospital stockholders. This is not to say that increased use of debt is not without risk and this should be recognized, particularly in an era of steadily increasing interest rates.

The ninth variable on our list was ‘uncompensated care as a percentage of net patient revenue’. For the purposes of our study, uncompensated care consists of bad debt charges plus financial assistance charges—which includes charity care. In our study, this variable was universally negatively associated with hospital bankruptcy. This ran contrary to our original expectations. One would logically think that as this variable increases, so would the likelihood of a poor financial outcome. However, we believe hospitals might be offsetting these costs via support from Medicaid disproportionate share (DSH) payments and financial support via charitable contributions, grants, tax exemption for not-for-profit organizations, and other external financing. Federal law requires Medicaid programs to make special payments to hospitals that serve a disproportionately large number of Medicaid and low-income patients. Without such payments, we believe this variable would likely point in the opposite direction. Unfortunately, the Affordable Care Act called for a reduction in federal DSH allotments starting in FY 2014, with the expectation that more people will be insured; the cuts have been delayed several times, but are currently set to take effect in 2024.

The last variable we consider on our list is ‘Medicare as a percentage of the total payer mix’, finding that it is associated with a lower overall risk of bankruptcy. This also came as somewhat of a surprise to us given statements by the American Hospital Association and other hospital advocacy groups that Medicare reimbursement is insufficient to cover costs [30,31]. We contend that, despite this viewpoint, Medicare may be more supportive of stemming off bankruptcy due to its relatively timely reimbursement—as quick as 14 days in contrast to a median of up to 55 days across all insurance payers up to a high nearing 80 days in some studies [32,33].

The remaining variables on our list are no less important than the ones we have discussed above, but lack the magnitude of our top ten listed factors. Having said this, it is important to take note of those variables that remain, including Medicaid as a percentage of payer mix, facility age, and the serious complication rate—all of which are positively associated with hospital bankruptcy. The findings appear to align with recent prior literature [34,35,36].

### Limitations and Suggestions for Future Research

This study has some limitations. First, there may be other influences on bankruptcy that we did not capture in our study. This is apparent in our lower-than-desired accuracy values in each of our analytical models. One extension we might consider in the future is the individual patient’s condition at the point of admission and discharge. Clearly, this would require a more detailed and comprehensive dataset of patient-level data than we currently have access to, but we believe it would add substantive depth and quality to the study.

Second, we acknowledge that our current study does not include specifics pertaining to the demographics or socio-economics of each hospital’s local patient population. Although we have included proxies for these factors in our study (i.e., urban/rural, teaching/non-teaching, etc.), none of these are precise measures of these important factors as there is likely a connection to the financial health of the hospital. One simple line of thought is that hospitals located in areas with young families or areas with challenging economic conditions would be more likely to be financially troubled. This would likely be reflected in less robust commercial insurance coverage in addition to fewer opportunities for philanthropy and charitable donations.

A third limitation of our study is that bankruptcy filing is, in certain cases, an optional decision of an organization experiencing financial distress. We conjecture that there may be some hospitals in our study that encountered financial difficulty but did not file for bankruptcy. This may be a contributory factor to the misclassification of bankrupt hospitals we found. Future work may benefit from considering something other than a dichotomous outcome or evaluating stages of organizational financial distress as the dependent variable.

Fourth, this study uses a single cross-section of data from the year 2019, thus, we are not able to assess the strengthening or weakening of our variables over time. Clearly, this is an area for additional research, as the measures that influence bankruptcy are likely to fluctuate somewhat with changes in policy, macroeconomic trends, inflation, and other related factors—not to mention major economic shocks such as what was witnessed during the COVID-19 pandemic. Likewise, each of our sample hospitals are in various stages of financial health and distress. Thus, a longitudinal study would assist us in more accurately capturing the subtle nuances of the factors that undermine or strengthen hospital financial health.

Lastly, this is a study strictly focused on hospitals in the United States. We should note that the business and operational environment of domestic hospitals is far different than any other industry and certainly different than healthcare in almost any other developed nation—in terms of how hospitals generate revenue, their ownership structures, and regulatory requirements. In our judgement, there is not a more convoluted, regulated, or complex industry on the planet than the US hospital industry. However, given the important role that hospitals have in the business and healthcare delivery landscape, we consider this a topic worth sharing.

## 5. Conclusions

Bankruptcy is the unfortunate result of many businesses. In recent years, there has been an increased number of hospitals in the United States that have met this troubled ending. Any time a business closes, there is a loss of goods, services, and economic opportunity in the local job market. However, with the closure of a hospital comes the increased societal cost of poorer access to care and other supportive clinical services. Thus, gaining a deeper understanding of what contributes to this financial outcome is increasingly vital as our population grows, and ages, but our care delivery facilities continue to close their doors. Based on our exploratory analysis, we contend both sound financial structure as well as supportive accreditation and quality performance all meaningfully insulate an organization against long-term economic underperformance.

## Figures and Tables

**Figure 1 healthcare-11-00165-f001:**
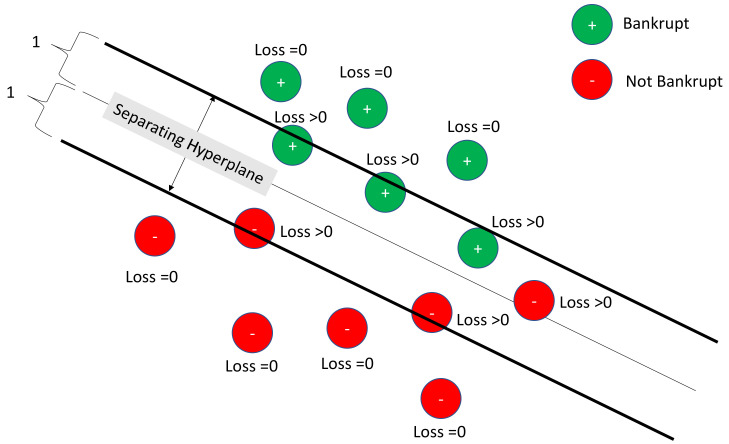
Example of a linear Support Vector Machine (hinge-loss).

**Figure 2 healthcare-11-00165-f002:**
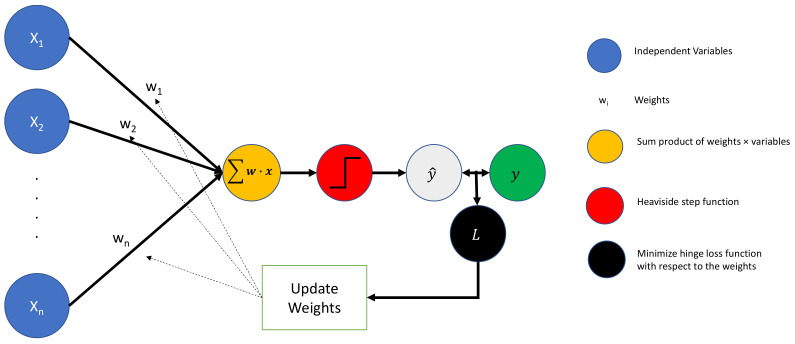
Perceptron model.

**Table 1 healthcare-11-00165-t001:** Descriptive Statistics.

Variable (N = 3121)	Mean (SD)	Min	Max
Bankrupt	0.009 (0.093)	0	1
Teaching Hospital	0.058 (0.233)	0	1
For Profit	0.248 (0.432)	0	1
Government	0.137 (0.344)	0	1
Sole Hospital	0.151 (0.358)	0	1
Market Concentration	0.337 (0.314)	0.020	1
Case Mix Index	1.635 (0.369)	0.710	5.260
Staffed Beds	192.475 (185.829)	1	2247
Urban	0.316 (0.465)	0	1
HCAHPS Recommend	0.706 (0.094)	0.270	1.000
TJC Certified	0.241 (0.428)	0	1
Medicare Percent	0.358 (0.13)	0	0.990
Medicaid Percent	0.092 (0.088)	0	0.820
Serious Comp Rate	0.99 (0.191)	0.460	4.370
Bed Utilization Rate	0.494 (0.203)	0	1
Facility Age	13.313 (9.425)	1.070	74.460
Current Ratio	6.278 (121.429)	−177.400	5102.400
Total Assets	$417,869,500 ($1,173,058,000)	($422,937,900)	$18,933,370,000
DCOH	46.768 (141.669)	−690.800	2318.000
Net Px Revenue	$299,155,500 ($436,603,700)	($61,287,150)	$5,951,047,000
Op Income per Bed	$5685.55 ($913,312.10)	($20,582,740)	$34,438,930
Accts Receivable	$104,934,300 ($198,282,000)	($82,540,520)	$3,711,121,000
Uncomp Care/Net Px Rev	0.08 (0.1)	−2.525	2.029
Debt/Equity	1.117 (21.456)	−179.250	741.950
Net Op Profit Margin	−0.026 (0.391)	−8.350	13.030
Asset Turnover	4257.932 (237,789.2)	−69.436	13,284,310.000

**Table 2 healthcare-11-00165-t002:** Logistic Regression, Sorted by Odds Ratios.

Variable	Coefficient (SE)	Odds Ratio (95% CI)	Z-Value (*p*-Value)
TJC Certified	−3.592 (0.453)	0.028 (0.011, 0.067)	−7.925 (<0.001)
HCAHPS Recommend	−3.518 (0.256)	0.030 (0.018, 0.049)	−13.744 (<0.001)
Accts Receivable	−2.978 (0.961)	0.051 (0.008, 0.335)	−3.097 (0.001)
Net Px Revenue	−2.787 (1.840)	0.062 (0.002, 2.267)	−1.515 (0.065)
Government	−2.477 (0.478)	0.084 (0.033, 0.214)	−5.18 (<0.001)
Current Ratio	−2.457 (1.025)	0.086 (0.011, 0.639)	−2.397 (0.008)
Uncomp Care/Net Px Rev	−1.921 (0.272)	0.146 (0.086, 0.249)	−7.065 (<0.001)
Staffed Beds	−1.812 (0.618)	0.163 (0.049, 0.549)	−2.929 (0.002)
Bed Utilization Rate	−1.793 (0.227)	0.166 (0.107, 0.260)	−7.892 (<0.001)
Debt/Equity	−1.625 (0.896)	0.197 (0.034, 1.140)	−1.813 (0.035)
Medicare Percent	−1.349 (0.190)	0.259 (0.179, 0.376)	−7.103 (<0.001)
Total Assets	−1.191 (2.928)	0.304 (0.001, 94.48)	−0.407 (0.342)
Market Concentration	−0.524 (0.173)	0.592 (0.422, 0.831)	−3.035 (0.001)
Case Mix Index	−0.467 (0.257)	0.627 (0.379, 1.038)	−1.813 (0.035)
Net Operating Profit Margin	−0.389 (0.345)	0.678 (0.344, 1.334)	−1.127 (0.130)
For Profit	−0.255 (0.176)	0.775 (0.549, 1.095)	−1.445 (0.074)
Urban	−0.084 (0.173)	0.920 (0.655, 1.291)	−0.483 (0.314)
Teaching Hospital	−0.007 (4.510)	0.993 (0.0, 6851.51)	−0.002 (0.499)
Op Income per Bed	0.115 (0.467)	1.122 (0.449, 2.803)	0.247 (0.403)
Sole Community Hospital	0.190 (0.130)	1.209 (0.937, 1.560)	1.458 (0.072)
Days Cash on Hand	0.209 (0.345)	1.233 (0.627, 2.424)	0.606 (0.272)
Asset Turnover Ratio	0.523 (0.225)	1.687 (1.087, 2.620)	2.330 (0.010)
Serious Complication Rate	0.856 (0.160)	2.355 (1.721, 3.222)	5.351 (<0.001)
Facility Age	0.904 (0.132)	2.468 (1.907, 3.195)	6.864 (<0.001)
Medicaid Percentage	1.160 (0.172)	3.190 (2.279, 4.465)	6.761 (<0.001)

Note: Highlighted lines identify variables that are statistically significant.

**Table 3 healthcare-11-00165-t003:** Performance Metrics of All Models.

Logistic Regression	Not Bankrupt	Bankrupt	Accuracy	Macro Average	Weighted Average
Precision	0.997159	0.021782	0.681614	0.509471	0.988411
Recall	0.680672	0.785714	0.681614	0.733193	0.681614
F1-Score	0.809066	0.042389	0.681614	0.425728	0.80219
Support	1547	14	0.681614	1561	1561
**SGD Classifier**	**Not Bankrupt**	**Bankrupt**	**Accuracy**	**Macro Average**	**Weighted Average**
Precision	0.997207	0.022587	0.693145	0.509897	0.988466
Recall	0.692308	0.785714	0.693145	0.739011	0.693145
F1-Score	0.817245	0.043912	0.693145	0.430579	0.81031
Support	1547	14	0.693145	1561	1561
**Neural Networks**	**Not Bankrupt**	**Bankrupt**	**Accuracy**	**Macro Average**	**Weighted Average**
Precision	0.997132	0.021359	0.675208	0.509246	0.988381
Recall	0.674208	0.785714	0.675208	0.729961	0.675208
F1-Score	0.804474	0.041588	0.675208	0.423031	0.797632
Support	1547	14	0.675208	1561	1561

**Table 4 healthcare-11-00165-t004:** Coefficient Comparison sorted by Average Absolute Value.

Variable	Logistic Regression	SGD Classifier	Neural Networks
Net Px Revenue	−2.78697	−6.90944	−24.13778
Accts Receivable	−2.97809	−6.12391	−24.28199
TJC Certified	−3.59153	−6.89102	−20.16941
Government	−2.47691	−4.96448	−20.97783
Current Ratio	−2.45735	−5.30466	−18.47199
HCAHPS Recommend	−3.51798	−3.85352	−16.17667
Total Assets	−1.19065	−3.16903	−11.00661
Debt/Equity	−1.62525	−2.06712	−11.63528
Uncomp Care/Net Px Rev	−1.92131	−1.92510	−10.87668
Medicare Percent	−1.34925	−1.54585	−9.96148
Staffed Beds	−1.81165	−1.26861	−6.46099
Bed Utilization Rate	−1.79318	−2.05771	−4.97037
Facility Age	0.90350	0.51885	6.75582
Medicaid Percent	1.15990	0.80188	4.77865
Case Mix Index	−0.46657	−0.00215	−5.87495
Serious Comp Rate	0.856396	1.08076	3.24159
Market Concentration	−0.52393	−0.80712	−1.24662
Net Op Profit Margin	−0.38921	−0.25601	−1.42210
For Profit	−0.25457	−0.72103	0.40809
Asset Turnover	0.52306	−1.16977	−33.78064
Op Income per Bed	0.11524	−0.80603	−2.47678
Urban	−0.08371	0.14662	3.04852
DCOH	0.20908	0.33661	−2.50597
Teaching Hospital	−0.00679	−2.25255	0.09292
Sole Hospital	0.18958	−0.13773	−1.58685

Note: Directionality differences are highlighted.

## Data Availability

Final models and all analyses were conducted in Python 3.7 and are posted to GitHub for readers’ review. They are available here: https://github.com/dustoff06/Bankruptcy/blob/main/Bankruptcy%20Final.ipynb.

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
