# Peer review of "The Predictive Factors of Hospital Bankruptcy—An Exploratory Study"

_healthcare, 2023, doi:10.3390/healthcare11020165_

Round 1
Reviewer 1 Report
The article offers a clear and rigorous approach to a relevant issue in bankruptcy prediction literature. The limitations of the study are clearly explained by the authors, both in the title itself and the body of the paper.
I have just a few minor issues I would like to address.
In the Discussion, you refer to “net patient revenue” and “accounts receivable” as the most important factors that can explain whether hospitals remain solvent. While the inclusion of net revenue in this list is rather obvious, the relevant role played in your models by accounts receivable is less intuitive. It is true, as you observe, that accounts receivable stem from sales, so they certainly indicate that revenues occurred in the past. However claiming that the presence of accounts receivable is a sign of a positive level of performance is debatable, because the issue here is liquidity, and increases in accounts receivable have direct negative effects on the hospital’s operating cash flow. A large amount of accounts receivable in itself could also signal difficulties in converting those sales into cash flows. As I understand, cash flows data are not available in your dataset. In case you extend the analysis to more years, it would be advisable to calculate them and include them in your model, or at least to calculate the change in net working capital, and see if it has any impact on your model.
Further in the same paragraph (line 359) you mention debt-to-equity ratio as a variable that signals reasonable use of debt financing, with the consequent tax savings and lower dividend distributions. It is certainly true that responsible use of financial leverage improves the hospital’s financial performance, but at the same time, it increases its financial risk, especially in case of sudden drops in operating profitability. So it seems rather counterintuitive that in your model the coefficient for this variable is negative. I think that this would merit further investigation in further studies; in the meantime, it would be useful to include these concerns in your discussion.
Finally, in line 38 you refer to Chapter 11 filings. If the article is intended for an international audience, it would be advisable to specify that it is a reference to U.S. bankruptcy law, with which not everybody is necessarily familiar outside of the United States.
Author Response
Reviewer 1, Comment 1: The article offers a clear and rigorous approach to a relevant issue in bankruptcy prediction literature. The limitations of the study are clearly explained by the authors, both in the title itself and the body of the paper. I have just a few minor issues I would like to address.
Author Comment 1 Response: Thank you for taking the time to review our work. Your insights and advice are invaluable to the peer review process.
Reviewer 1, Comment 2: In the Discussion, you refer to “net patient revenue” and “accounts receivable” as the most important factors that can explain whether hospitals remain solvent. While the inclusion of net revenue in this list is rather obvious, the relevant role played in your models by accounts receivable is less intuitive. It is true, as you observe, that accounts receivable stem from sales, so they certainly indicate that revenues occurred in the past. However claiming that the presence of accounts receivable is a sign of a positive level of performance is debatable, because the issue here is liquidity, and increases in accounts receivable have direct negative effects on the hospital’s operating cash flow. A large amount of accounts receivable in itself could also signal difficulties in converting those sales into cash flows. As I understand, cash flows data are not available in your dataset. In case you extend the analysis to more years, it would be advisable to calculate them and include them in your model, or at least to calculate the change in net working capital, and see if it has any impact on your model.
Author Comment 2 Response: Thank you for your comment. Yes, there is definitely a A/R to cash flow conversion process that cannot be overstated in its importance to economic sustainability to the contemporary health care organization. On page 11, lines 268-271 of the first draft, we allude to the importance of this process. In the revised draft we include language in the same area of the paper that states:
“Although there are clearly important nuances to how quickly receivables can be turned into cash flow, the simple existence of receivables balances on the balance sheet indicates a positive level of performance that is more likely than not to facilitate long term financial sustainability. Devoid of an opportunity to collect revenue – either directly or via a receivable – then the opportunity to generate cash flow ceases to exist. Further, given the specific characteristics of the third party payment system in the United States, it is virtually impossible to avoid carrying a receivables balance of some sort. Thus, we infer that a hospital showing a higher receivables balance is more likely one that is meeting patient demand and generating revenue. A simple truth in all businesses is revenue must exceed expenses. Healthcare is not immune from this fact.”
Reviewer 1, Comment 3: Further in the same paragraph (line 359) you mention debt-to-equity ratio as a variable that signals reasonable use of debt financing, with the consequent tax savings and lower dividend distributions. It is certainly true that responsible use of financial leverage improves the hospital’s financial performance, but at the same time, it increases its financial risk, especially in case of sudden drops in operating profitability. So it seems rather counterintuitive that in your model the coefficient for this variable is negative. I think that this would merit further investigation in further studies; in the meantime, it would be useful to include these concerns in your discussion.
Author Comment 3 Response: Thank you for this comment. We have added an additional line of text on page 12, line 309 that reads:
“This is not to say that increased use of debt is not without risk and this should be recognized, particularly in an era of steadily increasing interest rates.”
Reviewer 1, Comment 4: Finally, in line 38 you refer to Chapter 11 filings. If the article is intended for an international audience, it would be advisable to specify that it is a reference to U.S. bankruptcy law, with which not everybody is necessarily familiar outside of the United States.
Author Comment 4 Response: This is a good recommendation. We simply removed the “Chapter 11” reference and altered the wording to read “bankruptcy filings”.
Reviewer 2 Report
I found the work interesting.
I think the literature review should be extended and more recent works should be considered. The authors are dedicating a great amount of space to the Altman's score which is obsolete in the now-a-days dynamic and complex environment.
The authors are properly explaining how the model they have selected has been chosen and what are the datasources.
Please better explain the values of the variables in Table 2.
Please better explain how the proposed approach can be used in similar cases and what are the elements that makes it specially applicable to the medical/healthcare field.
Author Response
Reviewer 2, Comment 1: I found the work interesting.
Author Comment 1 Response: Thank you for taking the time to review our work. We are glad you found it interesting.
Reviewer 2, Comment 2: I think the literature review should be extended and more recent works should be considered. The authors are dedicating a great amount of space to the Altman's score which is obsolete in the now-a-days dynamic and complex environment.
Author Comment 2 Response: We view the work of Beaver (1966), Altman (1968), and Ohlson (1980) as seminal and sought to depart from the mainstream literature in our work given the context of the organizations in question. Little has been written recently on comprehensive studies of hospital bankruptcies.
Reviewer 2, Comment 3: The authors are properly explaining how the model they have selected has been chosen and what are the datasources.
Author Comment 3 Response: Thank you. No response appears to be expected for this comment.
Reviewer 2, Comment 4: Please better explain the values of the variables in Table 2.
Author Comment 4 Response: Based on your comment, we included the following:
“In the ‘Coefficient (Se)’ column of Table, the values of the coefficient (showing directionality and magnitude of the relationship between bankruptcy and the predictor in the “Variable” column) are provided along with the standard error, which is offset with parentheses. The ‘Odds Ratio (95% CI)’ column provides both the odds ratio (exponent of the coefficient) along with a 95% confidence interval in parenthesis. The associated Gaussian standard normal value along with the related p-value (in parentheses) is in the ‘Z-value (p-value)’ column. As an example, the variable ‘TJC Certified’ has a strong negative relationship with bankruptcy as indicated by the coefficient estimate and the standard error. Converting the coefficient to odds ratios shows that the odds of bankruptcy to no bankruptcy are 0.028 to 1.000, much smaller than 1:1 under the null hypothesis. The value in the standard normal of the coefficient is -7.925 or nearly 8 standard deviations below the mean. The p-value is therefore near zero indicating a statistically significant result.”
Reviewer 2, Comment 5: Please better explain how the proposed approach can be used in similar cases and what are the elements that makes it specially applicable to the medical/healthcare field.
Author Comment 5 Response: Based on your comment, we included the following.
“By using a collection of interpretable machine learning methods to estimate unseen, pristine test data, researchers can build models that provide coefficient directionality and effect size. These models can be compared for congruence based on coefficient magnitude and directionality. The better performing models may be identified and even ensembled (e.g., averaging) to push them forward for forecasting. In healthcare, this approach is particularly necessary to ensure that specious results from a single model are avoided and to handle the ‘big data’ that exists within the sector.”
Reviewer 3 Report
Using a series of indicators to predict the the bankcruptcy of organizations is usual in economic and management literatures, and relevant investigations have covered many industries. This study uses the hospitals as an example may be difficult to make sufficient theoratical advancement.
In regard of the soundness of methodology, this study applies data from the "Becker’s Hospital Review 2019". I am not sure why only the data of year 2019 is selected, while there can be a longitudinal dataset available. Besides, I am not sure whether there is multicolliniarity problem in the regression analysis, since the correlation matrix among variables has not been provided.
Author Response
Reviewer 3, Comment 1: Using a series of indicators to predict the the bankcruptcy of organizations is usual in economic and management literatures, and relevant investigations have covered many industries. This study uses the hospitals as an example may be difficult to make sufficient theoratical advancement.
Author Comment 1 Response: Thank you for your comments and kind consideration. In our review of the literature, there are very few extant studies of US hospital bankruptcies. And, to be honest, we are not seeking to advance theory. We desire to develop better understanding of the current US hospital environment with the intent of being able to assist health care leaders and policy makers to avoid future hospital closures and diminished access to care. We should also note that the business and operational environment of hospitals in the United States is far different than any other industry – both in terms of how hospitals generate revenue, but also in their ownership structures and regulatory requirements. In our judgement, there is not a more convoluted, regulated, or complex industry on the planet than the US hospital industry. Thus, given the important role that hospitals have in the US business and healthcare delivery landscape, we consider this a good topic to research.
Reviewer 3, Comment 2: In regard of the soundness of methodology, this study applies data from the "Becker’s Hospital Review 2019". I am not sure why only the data of year 2019 is selected, while there can be a longitudinal dataset available.
Author Comment 2 Response: As this is an exploratory study, a single year of data was our desired starting point for our investigation. To our knowledge, there is no comprehensive bankruptcy filing database available. Thus, all bankruptcies must be identified via literature searches of media and popular press announcement. With this comment in mind, we have since added the following sentence to the “Variables” section of the current draft.
“The year 2019 was specifically chosen as it is the most recent complete year of data devoid of any potential influence of the COVID-19 pandemic.”
Reviewer 3, Comment 3: Besides, I am not sure whether there is multicolliniarity problem in the regression analysis, since the correlation matrix among variables has not been provided.
Author Comment 3 Response: While we included the standard errors, a correlation matrix, and demonstrated coefficient stability across model categories, we did not specifically include a VIF table. That is now rectified and online at the GitHub site. The maximum VIF for the training data where we built the models is 4.4 (Net Patient Revenue) https://github.com/dustoff06/Bankruptcy/blob/main/Bankruptcy%20Final.ipynb.
Further, based on your comment, we included the following.
“For logistic regression on the training set (where the models were built), Variance Inflation Factors (VIF) were calculated. The highest VIF was 4.43 (Net Patient Revenue). Investigation of the model standard errors and coefficient directionality suggested that the logistic regression model provided stable and congruent results.”
Reviewer 4 Report
Dear authors,
Thank you for the opportunity to review your manuscript.
The main purpose of this manuscript is to develop a predictive model to guide the healthcare leaders towards understanding what specific factors can imperil hospitals and lead to their bankrupt. To achieve this purpose, the authors used data from the Definitive Healthcare and Becker’s Hospital Review 2019 (3,121 observations of 32 variables with 27 observed bankruptcies).
The methodology has sufficient detail to be replicable. The authors have used three models: 1) binary classification techniques that would produce parameter estimates with associated directionality and importance of the predictors including a logistic regression (LR) model; 2) linear support vector machine (SVM-stochastic gradient descent with a hinge loss function) model; and 3) perceptron (one-layer neural network-NN) model.
The conclusions are correlated with the results obtained, fully answering the research question. We can also mention the rich bibliography used, which includes relevant publications in the field.
In my opinion, the paper addresses a very interesting and topical issue, with a very interesting empirically analysis. The results of this analysis enrich the research on the hospital bankruptcy. However, I believe that some improvements should be implemented:
- in the Discussion section the authors must explain the results (the most important factors whether the hospital remains solvent) based on the literature; specific studies are mentioned only for some of the important factors; and
- the choice of 2019 for the study must be motivated. Why wasn't 2020 or 2021 chosen, to capture any particularities of the covid-19 period?
-should it be mentioned why a study on the US medical system would be of global interest? Which of the results might be useful for health care leaders in other countries to guide them in avoiding hospital bankruptcy?
Overall, I evaluate the study very positively and I recommend its publication after minor revision.
Author Response
Reviewer 4, Comment 1: Dear authors, Thank you for the opportunity to review your manuscript. The main purpose of this manuscript is to develop a predictive model to guide the healthcare leaders towards understanding what specific factors can imperil hospitals and lead to their bankrupt. To achieve this purpose, the authors used data from the Definitive Healthcare and Becker’s Hospital Review 2019 (3,121 observations of 32 variables with 27 observed bankruptcies).
Author Comment 1 Response: Thank you for taking the time out of your schedule to review our work!
Reviewer 4, Comment 2: The methodology has sufficient detail to be replicable. The authors have used three models: 1) binary classification techniques that would produce parameter estimates with associated directionality and importance of the predictors including a logistic regression (LR) model; 2) linear support vector machine (SVM-stochastic gradient descent with a hinge loss function) model; and 3) perceptron (one-layer neural network-NN) model. The conclusions are correlated with the results obtained, fully answering the research question. We can also mention the rich bibliography used, which includes relevant publications in the field.
Author Comment 2 Response: Thank you!
Reviewer 4, Comment 3: In my opinion, the paper addresses a very interesting and topical issue, with a very interesting empirically analysis. The results of this analysis enrich the research on the hospital bankruptcy. However, I believe that some improvements should be implemented. (First) in the Discussion section the authors must explain the results (the most important factors whether the hospital remains solvent) based on the literature; specific studies are mentioned only for some of the important factors; and
Author Comment 3 Response: In our review of the literature, there are very few extant studies of US hospital bankruptcies. That makes it difficult for us to explain our results via the literature. Given this is an exploratory study, we are really seeking to develop better understanding of the current US hospital environment with the intent of being able to assist health care leaders and policy makers to avoid future hospital closures and diminished access to care.
Reviewer 4, Comment 4: (second) the choice of 2019 for the study must be motivated. Why wasn't 2020 or 2021 chosen, to capture any particularities of the covid-19 period?
Author Comment 4 Response: Correct. We have since added the following sentence to the “Variables” section of the current draft.
“The year 2019 was specifically chosen as it is the most recent complete year of data devoid of any potential influence of the COVID-19 pandemic.”
Reviewer 4, Comment 5: (third) should it be mentioned why a study on the US medical system would be of global interest? Which of the results might be useful for health care leaders in other countries to guide them in avoiding hospital bankruptcy?
Author Comment 5 Response: Thank you for this thought. With this in mind, we have added the following language to the revised draft:
“Lastly, this is a study strictly focused on hospitals in the United States. We should note that the business and operational environment of domestic hospitals is far different than any other industry and certainly different than healthcare in almost any other developed nation – in terms of how hospitals generate revenue, their ownership structures, and regulatory requirements. In our judgement, there is not a more convoluted, regulated, or complex industry on the planet than the US hospital industry. However, given the important role that hospitals have in the business and healthcare delivery landscape, we consider this a topic worth sharing.”
Reviewer 4, Comment 6: Overall, I evaluate the study very positively and I recommend its publication after minor revision.
Author Comment 6 Response: Thank you again for your time and kind consideration.
Round 2
Reviewer 2 Report
Thank you for revised version. I have no further comments.
Author Response
We appreciate your time and contributions to our article!
Reviewer 3 Report
(1) Dear authors, the present revised manuscript seems a simplified version of business and financial research. At least from the reference part, I have not seen this manuscript sufficiently relevant to the healthcare management. I would like to suggest a series of citations which can enhance the relevance of this article to the field of healthcare management. I look forward to seeing a better revised manuscript.
Sakanga, V. I., Chastain, P. S., McGlasson, K. L., Kaiser, J. L., Bwalya, M., Mwansa, M., ... & Vian, T. Building financial management capacity for community ownership of development initiatives in rural Zambia. The International Journal of Health Planning and Management, 2020;. 35(1), 36-51.
Afriyie, S. O., Kong, Y., Lartey, P. Y., Kaodui, L., Bediako, I. A., Wu, W., & Kyeremateng, P. H. Financial performance of hospitals: A critical obligation of corporate governance dimensions. The International Journal of Health Planning and Management, 2020; 35(6), 1468-1485.
, , , , . Thailand's COVID-19: how public financial management facilitated effective and accountable health sector responses. The International Journal of Health Planning and Management, 2022; 37(4): 1894- 1906.
, , . Financial sustainability strategies of public primary health care centres in the Republic of Srpska, Bosnia and Herzegovina. Int J Health Plann Mgmt. 2021; 36( 5): 1772- 1788.
Kilci, E.N. A study on financial sustainability of healthcare indicators for Turkey under the health transformation program. Int J Health Plann Mgmt, 2021; 36: 1287-1307.
Karami Matin, B., Soltani, S., Byford, S., Soofi, M., Rezaei, S., Kazemi-Karyani, A., Hosseini, E. and Tolouei Rakhshan, S. The impacts of economic sanctions on the performance of hospitals in Iran: implications for human rights, International Journal of Human Rights in Healthcare, (2022), https://doi.org/10.1108/IJHRH-07-2021-0151
Malekzadeh, R., Yaghoubian, S., Hasanpoor, E. and Ghasemi, M. Health system responsiveness in Iran: a cross-sectional study in hospitals of Mazandaran province, International Journal of Human Rights in Healthcare, 2021; 14(2), 133-142. https://doi.org/10.1108/IJHRH-03-2020-0018
Asadi, H., Barati, O., Garavand, A., Joyani, Y., Kahkesh, M.B., Afsarimanesh, N., Seifi, M. and Shokri, A. Challenges facing hospital human resources during the COVID-19 pandemic: a qualitative study in Iran, International Journal of Human Rights in Healthcare, 2022;15(5), 489-498. https://doi.org/10.1108/IJHRH-03-2022-0016
(2) Besides, the highly imballanced dataset has only 27 observations recoded bankruptcy among 3127 observations. Whether the data structure would result in some bias in prediction is doubtful.
(3) Further, I am not sure how the proportion of observations of training set is determined. If the training set account for a large proportion of the total dataset, would such prediction using this method still has cost-effectiveness?
Author Response
Reviewer 3, Comment 1: Dear authors, the present revised manuscript seems a simplified version of business and financial research. At least from the reference part, I have not seen this manuscript sufficiently relevant to the healthcare management. I would like to suggest a series of citations which can enhance the relevance of this article to the field of healthcare management. I look forward to seeing a better revised manuscript.
- Sakanga, V. I., Chastain, P. S., McGlasson, K. L., Kaiser, J. L., Bwalya, M., Mwansa, M., ... & Vian, T. Building financial management capacity for community ownership of development initiatives in rural Zambia. The International Journal of Health Planning and Management, 2020;. 35(1), 36-51.
- Afriyie, S. O., Kong, Y., Lartey, P. Y., Kaodui, L., Bediako, I. A., Wu, W., & Kyeremateng, P. H. Financial performance of hospitals: A critical obligation of corporate governance dimensions. The International Journal of Health Planning and Management, 2020; 35(6), 1468-1485.
- Sachdev, S, Viriyathorn, S, Chotchoungchatchai, S, Patcharanarumol, W, Tangcharoensathien, V. Thailand's COVID-19: how public financial management facilitated effective and accountable health sector responses. The International Journal of Health Planning and Management, 2022; 37(4): 1894- 1906.
- Rakic, S, Djudurovic, A, Antonic, D. Financial sustainability strategies of public primary health care centres in the Republic of Srpska, Bosnia and Herzegovina. Int J Health Plann Mgmt. 2021; 36( 5): 1772- 1788.
- Kilci, E.N. A study on financial sustainability of healthcare indicators for Turkey under the health transformation program. Int J Health Plann Mgmt, 2021; 36: 1287-1307.
- Karami Matin, B., Soltani, S., Byford, S., Soofi, M., Rezaei, S., Kazemi-Karyani, A., Hosseini, E. and Tolouei Rakhshan, S. The impacts of economic sanctions on the performance of hospitals in Iran: implications for human rights, International Journal of Human Rights in Healthcare, (2022), https://doi.org/10.1108/IJHRH-07-2021-0151
- Malekzadeh, R., Yaghoubian, S., Hasanpoor, E. and Ghasemi, M. Health system responsiveness in Iran: a cross-sectional study in hospitals of Mazandaran province, International Journal of Human Rights in Healthcare, 2021; 14(2), 133-142. https://doi.org/10.1108/IJHRH-03-2020-0018
- Asadi, H., Barati, O., Garavand, A., Joyani, Y., Kahkesh, M.B., Afsarimanesh, N., Seifi, M. and Shokri, A. Challenges facing hospital human resources during the COVID-19 pandemic: a qualitative study in Iran, International Journal of Human Rights in Healthcare, 2022;15(5), 489-498. https://doi.org/10.1108/IJHRH-03-2022-0016
Author Response, Comment 1: Thank you for your suggestions. However, as this is an exploratory study of short-term acute care hospitals in the United States and given the distinctive nature of the US hospital industry among industrialized nations, we perceive the context of our study is dissimilar to any of those indicated in the suggested literature. The legal, policy, for-profit / not-for-profit ownership, and governmental / insurance industry financing of US hospitals are all unique factors when compared across an international perspective. Further, the primary focus of our study is pertaining to the phenomenon of hospital bankruptcy, not on the broad set of topics within the suggested articles (i.e., development initiatives in Zambia, public health centers in Sprpska, Bosnia and Herzegovina, the influence of corporate governance, the Turkish health system reforms, health system responsiveness in Iran, the impact of economic sanctions in Iran, or the specific influence of COVID-19, etc). Therefore the context of our supportive literature is constrained to bankruptcy related studies based on a US domestic context.
(2) Besides, the highly imballanced dataset has only 27 observations recoded bankruptcy among 3127 observations. Whether the data structure would result in some bias in prediction is doubtful.
Author Response, Comment 2: Thank you for the comment. As we discuss in the text, the data are highly imbalanced, which resulted in the use of a 50% training set augmented with 120 bootstrapped positive cases (see the Python code). The test set was left pristine and consisted of 14 bankruptcies. Forecasts from the training set were applied to these unseen and pristine test cases with the resulting metrics reported and Python code provided online. While the models are not perfect (e.g. 78% sensitive) but are 1) congruent and 2) largely directionally consistent, which is exactly what is needed for interpretable machine learning models identifying factors that may predict bankruptcy. Enclosed is the language from our study that explains this point:
“We then randomly selected a 50% test set which was separated from the data for use in prediction. Data were highly imbalanced with only 27 observations of bankrupt organizations in the set of 3,121 observations, so it was necessary to use an even split between the training and test sets. The final structure of the training set was 3,120 observations, whereas the test set was 1,561 observations. The test set included 14 bankrupt observations. Standard scaling (Z-transformation) was fit to the training set and the saved parameters from the training set were then used to transform the test set to ensure no information leakage. This transformation supported non-tree based methods which are not scale invariant. Due to the imbalanced data, the primary metrics of interest were precision (positive predictive value) and recall (sensitivity) as applied to the pristine test set. The F1 score, a weighted combination of precision and recall, as well as accuracy and specificity were evaluated in all cases. Where appropriate, parameter estimates and feature importance were calculated to estimate the effects of the independent variables. Hyperparameter tuning for all models was conducted using the training set data. Final models were then used to forecast bankruptcy status on the test set.”
(3) Further, I am not sure how the proportion of observations of training set is determined. If the training set account for a large proportion of the total dataset, would such prediction using this method still has cost-effectiveness?
Author Response, Comment 3: Thank you for this comment as well. As specified in the paper, the training set is 50% of the total observations with bootstrapped augmentation of the positive cases to improve model performance in classifying positive cases. While it would be nice to have more observations to use (e.g., 70 or 80%) in the training set, that is not feasible when so few positive observations exist. We call that out in the text (see below).
“Data were highly imbalanced with only 27 observations of bankrupt organizations in the set of 3,121 observations, so it was necessary to use an even split between the training and test sets.”
As you know, machine learning techniques typically rely on a training set (possibly broken into a validation piece for model building) and a pristine, undisturbed test set. Since the random split of the training and test set resulted in 14 observations in the test set, we had sufficient observations upon which our metrics are calculated.
We are not sure what you mean by ‘cost-effectiveness’ in the case of the train / test split, but it is computationally negligible to run this analysis in Python.
Thank you once again for taking the time to review our work.